# A Tunable Nanoplatform of Nanogold Functionalised with Angiogenin Peptides for Anti-Angiogenic Therapy of Brain Tumours

**DOI:** 10.3390/cancers11091322

**Published:** 2019-09-06

**Authors:** Irina Naletova, Lorena Maria Cucci, Floriana D’Angeli, Carmelina Daniela Anfuso, Antonio Magrì, Diego La Mendola, Gabriella Lupo, Cristina Satriano

**Affiliations:** 1Consorzio Interuniversitario di Ricerca in Chimica dei Metalli nei Sistemi Biologici (CIRCMSB), Via Celso Ulpiani 27, I-70126 Bari, Italy; irina_naletova@yahoo.com; 2Hybrid NanobioInterfaces Lab (NHIL), Department of Chemical Sciences, University of Catania, Viale Andrea Doria 6, I-95125 Catania, Italy; lorena.cucci@unict.it; 3Department of Biomedical and Biotechnological Sciences, University of Catania, Via Santa Sofia 89, I-95123 Catania, Italy; fdangeli@unict.it (F.D.); daniela.anfuso@unict.it (C.D.A.); 4Institute of Crystallography Catania, National Council of Research (IC-CNR), Via Paolo Gaifami 18, I-95126 Catania, Italy; leotony@unict.it; 5Department of Pharmacy, University of Pisa, via Bonanno Pisano 6, I-56126 Pisa, Italy

**Keywords:** plasmonics, nanomedicine, theranostics, copper, VEGF, glioblastoma, differentiated neuroblastoma, peptidomimetics, real-time quantitative polymerase chain reaction (qPCR), actin

## Abstract

Angiogenin (ANG), an endogenous protein that plays a key role in cell growth and survival, has been scrutinised here as promising nanomedicine tool for the modulation of pro-/anti-angiogenic processes in brain cancer therapy. Specifically, peptide fragments from the putative cell membrane binding domain (residues 60–68) of the protein were used in this study to obtain peptide-functionalised spherical gold nanoparticles (AuNPs) of about 10 nm and 30 nm in optical and hydrodynamic size, respectively. Different hybrid biointerfaces were fabricated by peptide physical adsorption (Ang_60–68_) or chemisorption (the cysteine analogous Ang_60–68_Cys) at the metal nanoparticle surface, and cellular assays were performed in the comparison with ANG-functionalised AuNPs. Cellular treatments were performed both in basal and in copper-supplemented cell culture medium, to scrutinise the synergic effect of the metal, which is another known angiogenic factor. Two brain cell lines were investigated in parallel, namely tumour glioblastoma (A172) and neuron-like differentiated neuroblastoma (d-SH-SY5Y). Results on cell viability/proliferation, cytoskeleton actin, angiogenin translocation and vascular endothelial growth factor (VEGF) release pointed to the promising potentialities of the developed systems as anti-angiogenic tunable nanoplaftforms in cancer cells treatment.

## 1. Introduction

In recent decades, protein-nanoparticle and peptide-nanoparticle conjugates have emerged as powerful nanomedicine tools, enabling biomedical applications in the prevention, diagnosis and treatment of disease [1,2]. Unfunctionalised, bare nanoparticles (NPs), are often able to match several of the desired functions required by theranostic platforms, including the peculiar optical, electrical, magnetic properties of nanometer-sized materials [3], the tunable geometries and the tailored size and surface chemistry [4] and the intrinsic biological properties, such as anti-angiogenic nanogold [5,6] or antibacterial nanosilver [7,8]. Biological protein-based nanoparticles are advantageous in having biodegradability, bioavailability, and relatively low cost. Many protein nanoparticles, for instance naturally occurring protein cages such as ferritin, are easy to process and can be modified to achieve desired specifications such as size, morphology, and weight [9,10,11]. Natural product-based nanomedicine include, among the most common types of nanoparticles, polymeric micelles, solid lipid nanoparticles, liposomes, inorganic nanoparticles and dendrimers [3,12]. 

Each of these nanoparticles has its own advantages and disadvantages as drug delivery vehicle. Hybrid peptide- or protein-NP conjugates enable addressing many of the difficulties that arise as results of in vivo applications, replacing many materials that have a poor biocompatibility and have a negative impact on the environment. Specifically, both naturally derived and synthetic polypeptides may offer improved biocompatibility [13], targeted delivery [14] and prolonged lifetime before clearance, to ensure an efficient therapeutic action [1,15].

Angiogenin (ANG) is a secreted ribonuclease (also known as RNase 5), identified in media from cancer cells, but also present in normal tissues, such as plasma and amniotic fluid, and secreted from vascular endothelial cells, aortic smooth muscle cells, fibroblasts [16]. Angiogenin induces neovascularization by triggering cell migration, invasion, proliferation, and formation of tubular structures [16,17,18,19]. 

Physiologically, ANG is overexpressed during inflammation, exhibiting wound healing properties as well as microbicide activity and conferring host immunity [20]. However, uncontrolled activity of angiogenin is implicated in pathological processes. 

The protein was isolated for the first time from medium conditioned by a human adenocarcinoma cell line (HT-29) [21]. A high expression of angiogenin has been described in different types of cancers and to their malignant transformation [16], including gliomas that are brain tumours fast-growing, aggressive and with a poor prognosis [22]. 

ANG expression has been identified also in neurons and acts as a part of the secretome of endothelial progenitor cells (EPCs) [23,24]; the modulation of ANG and EPCs as repair-associated factors has been found in stroke patients and mouse models of rehabilitation after cerebral ischemia [25]. Mutations in the ANG gene have been characterized in amyotrophic lateral sclerosis (ALS) [26] and Parkinson’s disease (PD) [27]. Moreover, endogenous angiogenin levels are dramatically reduced in an alpha-synuclein mouse model of PD and exogenous angiogenin protects against cell loss in neurotoxin-based cellular models of PD [27]. Genetic studies revealed that angiogenin treatment delays motor dysfunction and motor neuron loss, also prolonging the survival in superoxide dismutase 1 (SOD1) mouse model of ALS [18]. 

In motor neurons, ANG can be upregulated by hypoxia thought the stimulation of ribosomal ribonucleic acid (rRNA) transcription of endothelial cells [28]. Such a process is critical for the cellular proliferation induced by other angiogenic proteins, including vascular endothelial growth factor (VEGF) [29]. While the predominant role of VEGF in the formation of new blood vessels is unquestioned, several recent studies demonstrate that VEGF also has trophic effects on neurons and glia in the central- (CNS) and peripheral- (PNS) nervous system, promoting neurogenesis, neuronal patterning, neuroprotection and glial growth [30]. Therefore, VEGF modulates neuronal health and nerve repair; and exogenous angiogenin delivery can be considered a promising tool of anti-angiogenic therapy for treating gliomas, where malignancy is highly related to angiogenesis.

Copper is an essential metal that plays a key role in the CNS development and function, and its dyshomeostasis is involved in many neurodegenerative diseases as Alzheimer’s disease (AD), PD and ALS [31,32]. Furthermore, copper is known to be a strong angiogenic factor, with metal serum levels raising in a wide variety of human cancers [33,34]. Noteworthy, copper increases the expression of ANG and regulates its intracellular localization [35]. Moreover, ANG binds copper ions and the metal interaction largely influences its interaction with endothelial cells as well as its angiogenic activity [36,37,38]. Taking into account the correlations between angiogenin protein and copper in physiological and pathological conditions of the brain, a promising pharmacological approach in brain tumours therapy is the use of ANG as molecular target, whose activity may be modulated by the presence of copper ions. 

As an alternative to protein-based drugs, peptides mimicking functional domains of the whole protein are becoming more relevant as drug candidates, to address problems exhibited by the protein in in vivo applications such as the additional effect or functions or binding sites for other ligands, the immunological clearance before reaching their target site [39,40]. 

Three domains of angiogenin have been demonstrated essential to the protein to explicate its biological activity, i.e.,: the catalytic site (involving His-13, Lys-40, and His-114 residues), the nuclear translocation sequence (encompassing residues 31–35, RRRGL); the putative cellular binding site (residues 60–68, KNGNPHREN) [41,42]. In previous works, peptide fragments encompassing such different domains of the protein have been synthesized and used as mimicking model of the whole protein [6,43,44]. In particular, hybrid nano-assemblies of gold nanoparticles (AuNPs) functionalized with different peptides encompassing the putative cellular binding site of the protein (Ang_60–68_) have been demonstrated able to maintain their activity on cytoskeleton actin reorganization in a tumour cell line of human neuroblastoma [44]. 

Here, we report on the investigation of AuNPs functionalised with the peptide Ang_60–68_ or its analogous having a cysteine residue in the C-terminus (Ang_60–68_Cys), in the comparison with the whole ANG protein. Such systems have been scrutinised as potential nanomedicine platforms towards a brain cancer of human glioblastoma (A172 cell line). To compare the response of tumour and non-tumour brain cells, differentiated neuroblastoma (d-SH-SY5Y line) have been included in the study as model neuron-like cells.

Effects on cell proliferation, cytoskeleton actin changes, angiogenin translocation and VEGF expression upon the cell treatments with peptides- or protein-functionalized nanoparticles, in the absence or presence of copper ions, shed new light in the link between different factors involved in angiogenesis processes of non-tumour and tumour model brain cell cultures. Indeed, since ANG plays a key role in cell growth and survival, and the role of VEGF in brain tumour angiogenesis has been demonstrated [45], new perspectives in the therapeutic approaches may rely on the tiny modulation of the pro-/anti-angiogenic processes [46] for brain cancer treatment.

## 2. Results

### 2.1. Physicochemical Characterisation of Hybrid Peptides- and Protein-NP Conjugates

#### 2.1.1. Optical (Plasmonic) Properties Changes of AuNP upon Interaction with Peptides/Protein 

Peptide-functionalised NPs were fabricated by two different approaches, namely a purely physical adsorption and a prevalent covalent grafting, with the peptide sequences Ang_60–68_ and Ang_60–68_Cys, respectively. As positive control, samples of ANG-functionalised AuNPs were prepared by using the whole protein. 

Figure 1 shows the Ultraviolet (UV)−visible spectra of gold nanoparticles, before and after the addition of Ang_60–68n_, Ang_60–68_Cys or ANG, respectively. The plasmon peak parameters, i.e., the wavelength at the maximum absorbance (*λ_max_* = 519 nm) and the full width at half maximum (FWHM = 54 nm) point to the formation of a gold colloidal solution of spherical nanoparticles with an optical diameter of 11 nm [47]. 

The addition of 3 × 10^−5^ M Ang_60–68_ (Figure 1a) or Ang_60–68_Cys (Figure 1b) induced comparable red-shifts (Δ*λ_max_* = 3 nm) and hyperchromic-shifts (Δ*Abs*~0.07), according to previous findings [44]. No significant changes in the width of the plasmon peak were detected. The addition of the 1 × 10^−7^ M ANG whole protein (Figure 1c), lead to a significantly larger red-shift in the plasmon peak (Δ*λ_max_* = 4 nm) with respect to the bare nanoparticles than those found upon the addition of the peptides. Moreover, a hypochromic-shift (Δ*Abs* = −0.09) in comparison to the bare AuNPs and a broadening of the plasmon band (Δ*FHWM* = 11 nm) with the appearance of a shoulder at around 600 nm, were found, likely due to a partial nanoparticle aggregation.

As to the hybrid systems used for the cellular experiments, Figure 1d shows the UV-visible spectra of the protein/peptide-nanoparticle pellets samples after two washing steps, performed to remove unbound and/or weakly bound biomolecules.

The red-shift in the plasmon peak with respect to the bare AuNPs is still visible (Δ*λ_max_*~3 nm) for Ang_60–68_ and Ang_60–68_Cys peptides as well as for ANG protein). This finding confirms the irreversible adsorption of the peptides and protein molecules and hence the successful surface functionalisation of the gold nanoparticles by the used biomolecules.

#### 2.1.2. Hydrodynamic Size and Conformational Features of Peptides- and Protein-Functionalised NPs in the Absence or Presence of Copper Ions

To gain insight into the actual hydrodynamic size of the angiogenin-functionalised nanoparticles, dynamic light scattering was used to take into account the dynamic soft shell made by the peptide/protein molecules at the AuNP surface, in contrast to the optical size that reflects merely the ‘dry state’ or internal ‘core’ of the functionalised particle. The hydrodynamic size of ~30 nm for the aqueous dispersion of gold nanoparticles (1.7 × 10^8^ NP/mL) did not change significantly upon addition of the peptide solution (3 × 10^−5^ M), for both Ang_60–68_ and Ang_60–68_Cys fragments (Table 1). On the other hand, by addition of ANG (1 × 10^−7^ M), the nanoparticle size was largely increased in comparison to the bare AuNP, suggesting bridged interactions between the protein molecules immobilised at the nanoparticle surface that could also prompt a partial aggregation.

The Ang_60–68__NP and Ang_60–68_Cys_NP pellets maintained a size range comparable to that of non-rinsed nanoparticles (both bare and functionalized), while a slight decrease in the size was found for ANG_NP, where a fraction of loosely bound proteins molecules was therefore likely rinsed off by the washing steps of the protein-nanoparticle hybrids. 

Noteworthy, after the addition of copper ions, the average dimension of nanoparticles was still unchanged for bare AuNP, instead a dramatic increase in the hydrodynamic diameter was found for Ang_60–68__NP (to ~0.18 μm) and Ang_60–68_Cys_NP (to ~0.3 μm), respectively.

The Ang_60–68_ peptide is able to bind copper at physiological pH by the involvement of one imidazole, two deprotonated amide nitrogen and one carboxyl oxygen atoms, respectively [48]. The UV-vis parameters measured for the equimolar solutions at pH = 7.4 of copper(II) and Ang_60–68_Cys (*λ**_max_* = 624 nm; ε = 100 M^−1^cm^−1^) were very similar to those of analogous complex formed with Ang_60–68_ (*λ**_max_* = 630 nm; ε = 120 M^−1^cm^−1^).

Accordingly, the circular dichroism (CD) spectra of both Ang_60–68_+Cu(II) and Ang_60–68_Cys+Cu(II) (Figure 2) showed a minimum around 600 nm, assigned to copper d-d transition, and a broad band with a maximum approximately at 350 nm, assigned to charge transfer to the metal ion by the imidazole nitrogen (N_im_→Cu(II)) and the deprotonated amide nitrogen (N_amide_→Cu(II)).

### 2.2. Biological Characterisation of the Interaction between Peptides- or Protein-NP Conjugates and Brain Tumour (A172 line) or Non-Tumour (d-SH-SY5Y) Cells

#### 2.2.1. Determination of Angiogenin Expression in Glioblastoma (A172), Undifferentiated and Differentiated Neuroblastoma (SH-SY5Y) Cell Lines

To analyse the endogenous levels of ANG expression in the tested cancer cells (glioblastoma A172 and neuroblastoma SH-SY5Y) and neuronal-like cells (differentiated neuroblastoma, d-SH-SY5Y), we performed western blot analyses of protein extracts from crude cell lysates (Appendix A). Results confirmed that in tumour cells the expressed level of protein was significantly higher than in differentiated neuroblastoma (Appendix A). Moreover, to control the specific interaction of anti-angiogenin antibody with Ang_60–68_ or Ang_60–68_Cys in the comparison with ANG, the peptides and protein samples were analysed by Western and dot blotting assays. The used anti-angiogenin antibody detected only the whole protein but did not interact with the two peptide fragments (Appendix A).

#### 2.2.2. Cell Viability

MTT (3-(4, 5-dimethylthiazolyl-2)-2, 5-diphenyltetrazolium bromide) assays were carried out to assess the effect on cell viability (Figure 3) of peptides- or protein-functionalised NPs, in the absence or presence of copper ions, for brain glioblastoma (A172 line) and differentiated neuroblastoma (d-SH-SY5Y), respectively. 

In the tumour A172 cell line (Figure 3a), a significant increase on viability (+25%; *p* ≤ 0.05 vs. control untreated cells) was found after the treatment with ANG, both in absence and in the presence of added Cu(II). The cells incubation either with free peptides of Ang_60–68_Cys or Ang_60–68_ did not induce any significant change on cell viability; similar results were found for cells treated with peptides in the presence of copper. As to nanoparticle-treated cells, the incubation with bare AuNP, both in the absence and with Cu(II), did not modify the cell viability in comparison with control cells. The incubation with Ang_60–68__NP reduced the cell viability by about 20–25% (*p* ≤ 0.05 vs. the respective peptide and peptide + Cu(II) controls), both in the absence and in the presence of copper. As to Ang_60–68_Cys_NP, a significant cell viability decrease in the absence of Cu(II) (−25%; *p* ≤ 0.05 vs. the respective free peptide) was nullified by the incubation in presence of copper. A similar trend was found for the cells incubated with ANG_NP, where a reduced cell viability (−20%; *p* ≤ 0.05 vs. the respective free protein) in the absence of copper but no significant difference in presence of copper were found. 

In the non-tumour d-SH-SY5Y cell line (Figure 3b), none of the treatments used resulted in a statistically significant decrease of cell viability in comparison to untreated control cells. Trypan blue staining confirmed the above reported results (data not shown).

#### 2.2.3. Cytoskeleton Actin Reorganisation and Intranuclear Angiogenin 

Cell migration is a critical step in tumour invasion and metastasis; the regulation of this process is often monitored in therapies for treating cancer. Reorganization of the actin cytoskeleton is the primary mechanism of cell motility and is essential for most types of cell migration [49]. 

Confocal laser scanning microscopy (LSM) demonstrated substantial differences between the tumour A172 (Figure 4) and non-tumour d-SH-SY5Y (Figure 5) cell lines in the organization of the actin cytoskeleton, both before and after the treatments with the peptides/protein-conjugated nanoparticles, as well as the incubation in the copper-supplemented medium. 

Combined staining for F-actin (green) and nuclei (blue) for untreated glioblastoma (Figure 4, panel 1, CTRL) clearly shows their polygonal shape along with different types of actin dorsal fibres and transverse arcs, typical for the lamellipodial actin meshwork [50]. The cell treatment with bare AuNP and/or the addition of Cu(II) (Figure 4, panels 2–4) increased actin stress fibres. In contrast to A172 cells, F-actin staining of untreated d-SH-SY5Y cells (Figure 5, panel 1, CTRL) analysed by Laser Scanning Confocal Microscopy (LSM) showed several distinct types of actin structures with broad leading edges, including a lamellipodium with a loose meshwork of actin filaments, an actin rich lamella, dorsal ruffles, transverse arcs and stress fibres, as expected for differentiated neuroblastoma [51].

A172 cells treated with the free peptides or protein showed a diffuse actin staining for several lamellipodia protruding from the cell body in all directions. The addition of copper did not change significantly the actin staining for Ang_60–68_ (Figure 4, see panels 1 and 3), instead visibly decreased the lamellipodia structures for Ang_60–68_Cys (Figure 4, panels 1 and 3) and ANG (Figure 4, panels 1 and 3), respectively. Both in absence and in the presence of copper, A172 cells treated with Ang_60–68__NP (Figure 4, panels 2 and 4) and ANG_NP (Figure 4, panels 2 and 4) still displayed a similar actin staining than those incubated with the free peptide or protein, respectively.

On the contrary, after incubation with Ang_60–68_Cys_NP, both in the absence and presence of copper (Figure 4, panels 2 and 4), cells contained very few, if any, actin stress fibres in the central regions and lamellipodia structures. 

For d-SH-SY5Y cells treated with the two peptides or the protein or their nanoparticle conjugates, irrespective of the incubation in copper-supplemented medium or not, the most notable change was a generally less dense meshwork of actin filaments after the treatment with Ang_60–68_ (Figure 5, panels 1–4) or Ang_60–68_Cys (Figure 5, panels 1–4). The central region of these cells contained neither ventral stress fibres nor dorsal ruffles or transverse arcs detectable by LSM. On the contrary, numerous and diffuse actin structures, as well as prominent actin stress fibres along the entire cell border were found for cells treated with ANG samples (Figure 5, panels 1–4).

LSM imaging of intracellular angiogenin in glioblastoma and differentiated neuroblastoma cells confirmed the western blot results (see Appendix A) that untreated non-tumour d-SHSY5Y cells (Figure 5, panel 5) showed lower levels of endogenous ANG in the cytoplasm and in the nucleus than untreated tumour A172 cells (Figure 4, panel 5).

Cells incubated with ANG for 2 hr exhibited a strong increase of the red staining, confirming the cellular uptake of exogenous angiogenin. In A172 cells the staining was especially enhanced in the presence of copper ions for the nuclear and perinuclear regions (Figure 4, panels 7,8). In d-SH-SY5Y cells, an increased red staining was visible in vesicles in the cytoplasm and in the neurites (Figure 5, panels 5–8), according to intracellular angiogenin localisation reported by Thiyagarajan et al. in similar neuronal cell lines [52]. We also observed the presence of intense punctuate structure of angiogenin in perinuclear and neurite regions, suggesting formation of resembling secretory granules. 

Noteworthily, A172 cells treated with the peptide fragments Ang_60–68_ (Figure 4, panels 5–8) or Ang_60–68_Cys (Figure 4, panels 5–8) showed a diffuse cytoplasmic staining and a weak staining in the nucleus, neurites and cell membrane. A negligible staining of nuclear angiogenin was found after cell incubation with peptide-conjugated nanoparticles in the presence of copper ions.

#### 2.2.4. VEGF Release and Synthesis

VEGF has been identified as the most important pro-angiogenic factor released by cancer cells and its concentration in the tissue of glioblastomas has been demonstrated significantly higher than that in normal brain [53]. Moreover, VEGF has a crucial role in neurogenesis, neuronal patterning, neuroprotection and glial growth [30,54]. Figure 6 shows the VEGF release after incubation for 24 h of tumour A172 cells and d-SH-SY5Y with peptides- or protein-conjugated NPs, in the absence or presence of copper ions. 

In A172 cells (Figure 6a) the treatments with copper alone increased the VEGF release by about 2.0 folds (*p* ≤ 0.05 vs. control untreated cells), confirming the relevant role of this cation in cancer progression [55].

The incubation of the cells with Ang_60–68_Cys or with Ang_60–68_ did not modify the VEGF release in comparison to control cells, both in the absence and in the presence of Cu(II) whereas the treatments with ANG or ANG + Cu(II) increased the VEGF release by about 2.3-fold (*p* ≤ 0.05 vs. control untreated cells).

The treatments with bare AuNP, both in the absence and in the presence of copper ions, did not significantly modify the VEGF release in comparison to control cells. Surprisingly, the incubation with AuNPs functionalized with peptide fragment Ang_60–68_Cys and Ang_60–68_, induced a significant reduction of VEGF release respectively by 27% and by 30%, in comparison to the corresponding control (free Ang_60–68_Cys and free Ang_60–68_). Moreover, further reduction of the release was found when the incubation with peptide fragment Ang_60–68_Cys was performed in presence of copper. No difference was found in VEGF release after treatment of A172 cells with ANG_NP in comparison to cells treated with free ANG.

The incubation of d-SH-SY5Y cells with ANG did not modulate the VEGF release, as well as with Ang_60–68_ and Ang_60–68_Cys, both in absence and in presence of copper, in comparison to the respective controls.

Differently from A172 cells, in non-tumour d-SH-SY5Y cells (Figure 6b), only the treatment with AuNP functionalized with ANG, both in the absence and in presence of copper, induced an increase of the VEGF release in comparison to untreated control cells. The incubation of control cells with copper did not modulate the VEGF release. The concentration of VEGF released by A172 and d-SH-SY5Y was 133 pg/mL ± 10.1 and 58 pg/mL ± 4.3, respectively.

These results were confirmed by determination of VEGF messenger RNA (mRNA) levels (Figure 7). In A172 cells (Figure 7a) the treatments with ANG significantly increased mRNA transcription, whereas ANG_NP induced a significant reduction of transcription in comparison with free ANG but with values higher than control cells. No differences were found after treatment with free Ang_60–68_, free Ang_60–68_Cys as well as bare NPs. On the other hand, the incubation with Ang_60–68__NP and Ang_60–68_Cys_NP induced a significant reduction of mRNA transcription by about 2.2 and 2.7 folds, respectively, in comparison to the respective control (free Ang_60–68_ and free Ang_60–68_ Cys). Moreover, further reduction of the transcription was found after incubation in the presence of copper.

In non-tumour d-SH-SY5Y cells (Figure 7b), only the treatment with AuNP functionalized with ANG, both in the absence and in presence of copper, induced an increase of the VEGF mRNA transcription in comparison to the respective control (free ANG). 

## 3. Discussion

In this study, two brain cell lines, namely tumour glioblastoma (A172) and differentiated neuroblastoma (d- SH-SY5Y) neuron-like cells, were scrutinised after incubation with hybrid nanoassemblies made of gold nanoparticles functionalised with angiogenin protein or with two different angiogenin-mimicking peptides (Ang_60–68_ and its cysteine derivative at the C-terminus, Ang_60–68_Cys) containing the ANG residues from 60 to 68, which is the the exposed protein loop region that is part of a cell-surface receptor binding site [56]. 

The Ang_60–68_ peptide has been demonstrated to specifically interact with cytoskeleton actin [6], whereas the Ang_60–68_Cys peptide has been successfully used to tailor gold nanoparticles by chemical grafting [44]. Gold, indeed, being a soft acid, binds to soft bases like thiols, to form stable Au-S bonds (40–50 kcal/mol) that are able to replace the citrate shell on the nanoparticle surface due to the strong affinity binding of the thiol groups with the metal [57]. 

In a previous study, we scrutinised the actual immobilisation of Ang_60–68_Cys and Ang_60–68_ onto the surface of AuNPs upon the simple addition of the peptide solution to nanoparticles dispersed in water. Indeed, by a multitechnique characterisation approach, including UV-visible, attenuated total reflectance–Fourier transform infrared (ATR/FTIR) and circular dichroism (CD) spectroscopies as well as atomic force microscopy(AFM), we could demonstrate an irreversible strong interaction between the peptide molecules and the gold nanoparticles resulting into the biomolecule-coated nanoparticles [44]. In the present work, to functionalize the gold nanoparticles with Ang_60–68_, Ang_60–68_Cys or ANG, the biomolecules were added to the colloidal dispersion (1.7 × 10^8^ AuNP/mL) at the concentration respectively of 3 × 10^−5^ M for the peptides and 1 × 10^−7^ M for the protein, and the shifts in the plasmon band were monitored (Figure 1).

The optical interface established between the biomolecules and the metal nanoparticle surface, as investigated by UV–visible spectroscopy, clearly evidenced an irreversible immobilisation of the peptides and protein molecules onto AuNPs (Figure 1d). 

Noteworthy, a red-shift in the wavelength of maximum absorption ($\lambda_{max}$) as well as a broadening in the FWHM of the plasmon peak were found for both peptides- and protein-added nanoparticles in comparison to bare AuNPs. These spectral changes point to an increase in the nanoparticle optical size, which is dependent on the following two concomitant processes: (*i*) nanoparticle surface decoration by biomolecules adsorption; (*ii*) nanoparticles aggregation. 

The latter contribution was most evident for the protein, as displayed by the plasmon peak broadening and the appearance of a shoulder approximately at 600 nm of wavelength. Hence, the NP functionalisation by the biomolecules immobilisation resulted in peptide-conjugated NPs with lower tendency to aggregation than the protein-conjugated NPs. To better understand these findings, the nanoparticle coverage (Γ, in molecule/NP) was calculated from the changes in $\lambda_{max}$ by using equations (1), (2) and (3). 

Theoretical predictions show how the local refractive index environment of a metal nanoparticle affects its absorption spectrum. By assuming the protein-coated nanoparticles as core-shell spheres with a metallic core of d diameter, corresponding to the uncoated nanoparticles, and a homogeneous spherical proteinaceous shell, the fraction of protein over the total particle, g, is related to the changes in the wavelength of maximum absorption for uncoated colloid ($\lambda_{max,0}$), as given by Equation (1):
(1)$$g=\left(1+\alpha_{s}\right){\left(\frac{\lambda_{p}^{2}\left(\varepsilon_{s}-\varepsilon_{m}\right)}{\Delta \lambda \cdot \lambda_{max,0}}+2\alpha_{s}\right)}^{-1},$$
where $\lambda_{p}$ is the free electron oscillation wavelength (which is 131 nm for gold [58]), ε is a dielectric constant or relative permittivity (equal to the squared refractive index); $\alpha_{s}=\frac{\left(\varepsilon_{s}-\varepsilon_{m}\right)}{\left(\varepsilon_{s}+2\varepsilon_{m}\right)}$ is the polarizability of a sphere with shell dielectric constant $\varepsilon_{s}$ in a surrounding medium of dielectric constant $\varepsilon_{m}$. According to Equation (2), which refers to the shell thickness (*s*):
(2)$$s=\frac{d}{2}\left[\frac{1}{{\left(1-g\right)}^{1/3}}-1\right],$$
and using the Feijter’s formula in Equation (3):
(3)$$\Gamma =s\frac{n_{s}-n_{m}}{dn/dc},$$
where Γ is the coverage and dn/dc is the refractive index increment (typically 0.19 mL⋅g^−1^ for a protein [59]), the mass of protein absorbed per unit area can be calculated.

The estimated values from the experimental spectroscopic data as well as the theoretical coverage calculated by considering an ideal monolayer in the two limit configurations respectively of end-on or side-on, are given in Table 2.

From Table 2 is evident that a multilayer coverage can be presumed for ANG_NP, while most likely a monolayer in end-on configuration and a sub-monolayer coverage can be assumed for Ang_60–68__NP and Ang_60–68_Cys_NP, respectively. Hence, in the case of ANG, many protein molecules adsorbed at the nanoparticle surface and formed a ‘thick’ shell that could perturb the mechanism of electrostatic stabilisation for colloidal gold [3], thus explaining the partial aggregation measured in UV-visible spectra. As to the two peptide fragments, their smaller size lead to the formation of a thinner and stiffer shell around the nanoparticles than that formed by the protein molecules. This picture is further supported by the coverage calculated for the pellets recovered after the washing steps. Indeed, for ANG_Au pellet, a loss of unbound and/or weakly bound proteins of about 78% can be estimated by the protein fraction shell decrease to g = 0.5, which corresponds to coating thickness and absorbed protein mass of s = 1.54 nm and Γ = 36 ng/cm^2^, respectively. On the other hand, for both Ang_60–68__NP and Ang_60–68_ Cys_NP pellets, the calculated values for the peptide shell are still g = 0.74, s = 3.36 nm and Γ = 79 ng/cm^2^. To note, even if the measured plasmon peak changes were comparable upon their approaching at the interface with the gold nanoparticles, the cysteine residue in Ang_60–68_Cys is expected to drive, through the thiol-gold bonding [44], a more ordered and compact biomolecule gathering at the nanoparticle surface in comparison to Ang_60–68_, which is in agreement with the estimation of a sub-monolayer coverage in Ang_60–68_Cys_NP. 

The DLS method is a reliable instrumental tool for non-perturbative and sensitive diagnostics of the aggregation processes of gold nanoparticle conjugates initiated by biospecific interactions on their surface [62]. Indeed, the nanoparticle hydrodynamic size determined by DLS (Table 1) showed the same trend of optical size change; moreover, also evidenced nanoparticles aggregation induced by the addition of copper ions to the peptides- or protein-functionalised NPs (i.e., hydrodynamic size increase approximately of 373%, 866% and 15% for Ang_60–68__NP, Ang_60–68_Cys_NP and ANG_NP, respectively) but no size change for the bare AuNPs.

Transition metals, such as copper, can prompt the aggregation of proteins and peptides through the formation of metal complexes [63]. It is known that ANG is able to bind to copper ions [43] and bridged copper complexes can lead to the formation of nanoparticles clusters. This effect was more evident for the hybrids CysAng_60–68__NP, where the prevalent chemisorption process leads to a more ordered arrangement of the biomolecules around the nanoparticles. 

The differences found in the presence of copper ions could also be due to different binding modes between the copper and the peptides- or the protein-functionalised nanoparticles. The UV-visible parameters of copper complexes formed by Ang_60–68_ and Ang_60–68_Cys were similar, suggesting that the metal ion experiences the same coordination environment with both peptides. However, the observed blue-shift (Δλ = 6 nm) and the parallel decrease of molar absorbance coefficient (Δε = 10) for Ang_60–68_Cys + Cu(II) compared to Ang_60–68_ + Cu(II), suggest a slight increase of ligand field strength and a more planar disposition of donor atoms bound to the metal ion [64,65]. 

The CD spectra (Figure 2) confirmed the involvement of imidazole and deprotonated amide nitrogen as donor atoms in metal binding for copper complexes formed by the two peptides [48]. The sharper peaks around 300 nm evidenced the slight increase of metal binding affinity of Ang_60–68_Cys. Furthermore, the CD broad band in the d-d transition region suggested that the extra cysteine residue at C-terminus may affect peptide backbone conformation of Ang_60–68_Cys more than it happens for Ang_60–68_+Cu(II) system [66]. As for the protein, the main copper anchoring sites are the RNase catalytic sites His-13 and His-114 [38]; therefore ANG displays a different coordination mode compared to copper complexes formed by Ang_60–68_ and Ang_60–68_Cys. However, it has been hypothesized that in the presence of excess copper a second metal ion can bind to the 60–68 region of ANG affecting protein binding with cell membrane [37]. The different metal coordination modes may potentially tune the biological response of functionalized nanoparticles.

The tests of cell viability/proliferation, cytoskeleton actin, angiogenin translocation and VEGF release were scrutinised both in basal and in copper-conditioned medium. Noteworthy, copper is another co-player of the angiogenesis process [33,34].

The cell response to nanoparticles is strongly dependent on the cell line, since, for instance, different cell models can overexpress different receptors at the membrane that may trigger the nanoparticle internalisation. ANG stimulates the expression of ANG receptors which mediate its nuclear translocation [16,28]; when nuclear translocation of ANG is inhibited, its angiogenic activity is abolished [67]. 

Neuroblastoma SH-SY5Y cells are used as a model of dopaminergic neurons as the cells possess similar biochemical functionalities of neurons. They are able to synthesize dopamine and also express dopamine transporter on the cell membrane [68]. On the other hand, SH-SY5Y cells have very low levels of the redox protein thioredoxin that together with glutathione redox cycle represents the major cellular redox buffer [69], acts as a growth factor and is found to be overexpressed in many human primary cancers including glioblastoma cells [70]. 

As to the cell viability effects measured on tumour glioblastoma (A172) and non-tumour differentiated neuroblastoma (d- SH-SY5Y) cell lines, our results (Figure 3) pointed to the very promising potentialities of peptide- and protein-functionalised gold nanoparticles to decrease the proliferation of tumour cells. 

Indeed, after 24 h of A172 cells incubation with Ang_60–68__NP, Ang_60–68_Cys_NP and ANG_NP a significantly decreased viability was found compared the cells treated with the free peptides or protein molecules as well as to the untreated control. Noteworthy, at the used experimental conditions, the bare AuNPs as well as the free peptides were found to not affect the cell viability, whereas the free protein increased the viability in comparison to untreated cells, both in the absence and in the presence of copper ions. Another interesting cue was found in the experiments performed in copper-supplemented medium, where cell treatments with Ang_60–68_Cys_NP+Cu(II) and ANG_NP + Cu(II) nullified the abovementioned decrease of cell viability, whereas for cells treatments with Ang_60–68__NP+ Cu(II) no significant differences were found with respect to Ang_60–68__NP. These findings confirmed the higher capability in the copper binding for the Ang_60–68_Cys- conjugated nanoparticles with respect to Ang_60–68_-NP, as discussed above from CD results. 

In contrast to A172 cells, no toxicity nor increase in viability was observed for any of the incubation conditions of differentiated neuroblastoma cells, as expected for not proliferating non-tumour cells. These findings further support the good potentialities of our peptides- and protein-conjugated NPs as cell specific, anti-angiogenic nanomedicine tools. 

Angiogenin is a protein with an extreme positive charge (pI >10.5), thus generally can avidly bind the cellular membrane [71]. Indeed, ANG binds to the membrane surface actin of vessel endothelial cells and activate the matrix protease cascades. 

In the cytosol, angiogenin encounters an endogenous inhibitor protein, known as ribonuclease inhibitor (RI), which binds to angiogenin to form a complex with a dissociation constant value in the low femtomolar range, stabilized largely by favourable Coulombic interactions, as RI is highly anionic [72]. It has been demonstrated that upregulating RI suppresses tumour growth and tumour microvessel density through suppression of ANG function [73]. 

In order to explain the different response of two neural lines, we analysed via confocal microscopy the remodelling of actin filaments as well as the ANG translocation induced by the peptides- or protein-conjugated NPs, both in the absence and in the presence of Cu(II) (Figure 4 and Figure 5).

Our results in tumour A172 cell line showed that the cell treatment with bare AuNPs and/or the addition of Cu(II) significantly increased actin stress fibres, while after incubation either with the free Ang_60–68_ peptide as well as its NP-conjugated derivative, an enhanced actin staining for several lamellipodia protruding from the cell body in all directions were found, with no significant changes observed in the presence of copper. A similar strong actin staining for lamellipodia in Ang_60–68_Cys peptide-and ANG protein-treated cells was instead decreased by the presence of copper. Finally, after incubation with Ang_60–68_Cys_NP, both in the absence and presence of copper, cells contained very few, if any, actin stress fibres in the central regions and lamellipodia structures. 

As to the non-tumour d-SHSY5Y cell line, no significant changes in the cytoskeleton actin were found for cell incubation in the presence or not of copper ions, but only a generally less dense actin meshwork after the treatment with Ang_60–68_ or Ang_60–68_Cys samples. Numerous and prominent actin stress fibres along the entire cell border were found for cells treated with ANG samples. 

As migration and, thus, infiltration of glioma cells is largely governed by reshaping the cytoskeleton, it is no surprise that the composition and organization of the cytoskeleton in glioma cells differs strongly from that of healthy brain cells, such as the neuron-like differentiated neuroblastoma cells. In a study on glioblastoma multiforme (GBM), the most lethal brain tumour, Memmel et al. found that inhibition of cell migration was associated with massive morphological changes and reorganization of the actin cytoskeleton [50]. 

ANG can interact with the actin, a protein able to form different polymeric structures inside the cells, which is essential to maintain the cell structure and motility [74]. The result of the binding to the actin is the inhibition of the polymerization with consequent changing of the cell cytoskeleton. These modifications play a fundamental role during proliferation of both endothelial and tumour cells [75]. The role of ANG in cell migration, necessary for tumour invasion and metastasis, has been confirmed by an important study which detected elevated levels of secreted and cell surface-bound ANG in highly invasive metastatic breast cancer cells. It has been indeed demonstrated that ANG interacts with the plasminogen activation system, thus increasing plasmin formation and cell migration of tumour cells [76]. 

Under physiologic conditions, ANG is present in the nucleus and in the cytoplasm, where is held in an inactive state through interaction with its known inhibitor Ribonuclease/Angiogenin Inhibitor 1 (RNH1), which prevents random cleavage of cellular RNA. A minor pool of ANG is secreted and is internalized by surrounding cells with a mechanism of endocytosis receptor mediated (reviewed by Shawn [71]. In stressed cells ANG dissociates from his inhibitor and becomes active. In this condition, nuclear ANG translocates from nuclear to the cytoplasmic compartment where cleaves mature transfer ribonucleic acid (tRNA), releasing two smaller RNA fragments, termed 5′- and 3′ tRNA-derived Stress-induced RNAs (tiRNAs). The post-transcriptional tRNA processing is necessary to allow the tRNA to regulate in specific manner the transcription [77]. Moreover, tRNA fragments can bind to cytochrome c and block the apoptosoma assembling, thereby inhibiting caspase-3, with consequent increasing of cell viability and proliferation [78]. Noteworthy, the nuclear concentration of ANG increases in the endothelial cells under the stimulation with basic fibroblast growth factor (bFGF), VEGF, acidic fibroblast growth factor (aFGF), epidermal growth factor (EGF), and foetal bovine serum (FBS). 

Authors hypothesized that the endogenous angiogenin participates in endothelial cell proliferation induced by other angiogenic factors [29]. Recombinant ANG plays an important role in neuroprotection against excitotoxic and endoplasmic reticulum (ER) stress in primary motor neuron cultures, and in SOD1G93A mice [18].

By LSM we were able to visualise the different intracellular localisation of endogenous angiogenin in A172 and d-SH-SY5Y cells, but similar effects on angiogenin translocation or uptake by the treatment with the peptides- or protein-conjugated nanoparticles, respectively.

For tumour glioblastoma, we found the endogenous angiogenin localised in the nucleus and in the cytosol, while neuron-like differentiated neuroblastoma displayed a weaker angiogenin staining (according to western blotting analysis, Appendix A), mainly localised in cytoplasm. Indeed, A172 cells treated with the free protein or the protein-conjugated NPs exhibited a strong angiogenin staining of the nuclear and perinuclear regions, especially for incubation in copper-supplemented medium. In d-SH-SY5Y cells, most of the protein was visible in the cytoplasm as large speckles but is also present in the nucleus, in the neurites and the membrane. In both cell lines, upon the treatment with bare AuNPs and/or the incubation in copper-supplemented medium, structural perturbation of intracellular angiogenin was observed, with an intense punctuate structure in perinuclear and neurite regions that suggested the formation of resembling secretory granules. 

The treatment with peptides or peptide-conjugated nanoparticles was able to translocate angiogenin, with a diffuse cytoplasmic staining and a weak staining in the nucleus, neurites and cell membrane after the incubation with Ang_60–68_ or Ang_60–68_Cys; in the presence of copper-supplemented medium, most of the protein remains in the cytoplasm and is absent from the neurites and membrane. 

Nuclear angiogenin plays different roles. It is localised inside the nucleolus, centre of synthesis and assembly of the ribosomes, where stimulates rRNA production, required for cellular proliferation [79]. Moreover, ANG bind the histone protein and cause a modification which regulates mRNA transcription. The ability to bind the DNA allows to ANG to function as a adaptor protein which recruit other modifying enzymes with methyltransferase or acetyltransferase activity [16]. Moreover, ANG has a nuclear localization sequence (NLS), containing Arg 33, which equips the protein for nuclear import [28]. After entering the nucleus, ANG accumulates in the nucleolus, which is the site of ribosome biogenesis. Within the nucleolus, ANG stimulates ribosomal DNA (rDNA) transcription [80].

Our results demonstrate that ANG was able to enter glioma cells and to induce their proliferation. In A172 cancer cells, the mitogen activated protein kinase (MAPK)/extracellular-signal-regulated kinase (ERK) signalling pathway could be responsible of the ANG phosphorylation which prevent the binding with the RI. Consequently, free ANG could exercise his effect in the nucleus, promoting DNA transcription and cell proliferation [81]. The effect of ANG on A172 cells, but not of ANG_NP, was highlighted by the increase of VEGF transcription and release, after the ribonuclease activity of the protein in the nucleus. 

The lower VEGF release after incubation of the cells with ANG_NP could be determined by the presence of NPs, which could prevent the phosphorylation of the protein, essential step for its nuclear translocation. A low concentration of phosphorylated protein could explain the reduction of VEGF release after incubation with ANG_NPs in comparison to free ANG.

The peptide fragments Ang_60–68_Cys and Ang_60–68_ were able to enter the cells but only few of them could cross nuclear membranes, because they are missing of the nucleolar targeting, specifically of Arg 33; the consequent effect is a level of release of VEGF very similar to control cells. Instead, Ang_60–68_Cys_NP and Ang_60–68__NP entered the nucleus and successively they could bind rDNA. Probably, the fragments were not able to catalyse the digestion of the RNA on the promotor site, due to the lack of the catalytic sequence. Consequently, the dissociation of the transcription termination factor I-interacting protein (TIP5) from the rDNA promoter did not occur. The ensuing steric obstruction by the binding of Ang_60–68_Cys_NP and Ang_60–68__NP to rDNA could block the binding of other native angiogenin molecules thereby significantly reducing rDNA transcription and VEGF production. The shield-effect of the NPs towards peptide fragments Ang_60–68_Cys and Ang_60–68_ could represent an interesting strategy to modulate VEGF release by glioma cells.

The different response of the SH-SY5Y cells to ANG could depend by the interaction of the peptide with his inhibitor, as the signalling pathway determining the phosphorylation are switch off in not tumour cells. In this condition, the protein could remain inactive in the cytosol, bound to its inhibitor.

In differentiated SH-SY5Y cells, the hybrid ANG_NP increased VEGF release in comparison to free ANG probably because in absence of NPs, after the adhesion to- and crossing- through plasma membrane, it binds its inhibitor inside the cells. The hybrid ANG_NP protects the protein by the bind with the inhibitor, thereby increasing VEGF mRNA transcription and VEGF release in comparison to free ANG probably because the presence of the NPs could protect the protein by the binding with the inhibitor, thereby increasing VEGF mRNA transcription and VEGF release. 

## 4. Materials and Methods 

### 4.1. Chemicals

Gold(III) chloride trihydrate (CAS Number 16961-25-4), trisodium citrate dihydrate (CAS Number: 6132-04-3), 3-(*N*-morpholino)propanesulfonic acid (MOPS, 1132-61-2), potassium chloride (7447-40-7), sodium chloride (7647-14-5), tris(2-carboxyethyl)phosphine (TCEP, 51805-45-9), hydrochloric acid (7647-01-0), nitric acid (7697-37-2), sodium hydroxide (1310-73-2), *N*,*N*-diisopropyl-ethylamine (DIEA, 7087-68-5), *N*,*N*-dimethylformamide (DMF, 68-12-2), 20% (*v*/*v*) piperidine (110-89-4) in DMF solution, *N*-hydroxybenzotriazole (HOBt, 123333-53-9), triisopropylsilane (TIS, 6485-79-6), trifluoroacetic acid (TFA, 76-05-1), isopropyl β-d-1-thiogalactopyranoside (IPTG, 367-93-1), Tris(hydroxymethyl)aminomethane hydrochloride (Tris-HCl buffer, 1185-53-1), ethylenediaminetetraacetic acid (EDTA, 60-00-4), guanidine hydrochloride (GdnHCl), 1,4-dithiothreitol (DTT, 3483-12-3), phosphate buffered saline (PBS) tablets and 3-(4,5-dimethyl-2-thiazolyl)-2,5-diphenyl-2H-tetrazolium bromide (MTT, 298-93-1), ethylene glycol-bis(2-aminoethylether)-*N*,*N*,*N*′,*N*′-tetraacetic acid (EGTA, 67-42-5), nonyl phenoxypolyethoxylethanol (NP40, 9016-45 -9), bovine serum albumin (BSA), 3,3′,5,5′-tetramethylbenzidine (TMB, 54827-17-7) sulphuric acid (7664-93-9) and Triton X-100 (9002-93-1) were purchased from Sigma-Aldrich (St. Louis, MO, USA). 2-(1-*H*-Benzotriazole-1-yl)-1,1,3,3-tetramethyluronium tetrafluoroborate (TBTU) was purchased from Novabiochem (Läufelfingen, Switzerland). 

The designed primers for angiogenin (ANG) protein expression were purchased from Eurofins GWM (Ebersberg, Germany). The over-expression plasmid (pET22b(+)-ANG), including a codon-optimized gene for ANG, was obtained from Sloning BioTechnology (Puchheim, Germany). Terrific Broth (TB) liquid microbial growth medium, Dulbecco’s modified eagle medium (DMEM), Ham’s F-12 medium (F12), streptomycin, l-glutamine, foetal bovine serum (FBS) were provided by Lonza (Verviers, Belgium). DMEM high glucose 30-2002 was provided by ATCC (LGC Standards S.r.l., Sesto San Giovanni (MI), Italy). Ultrapure MilliQ water was used (18.2 mΩ·cm at 25 °C, Millipore, (Burlington, MA, USA).

### 4.2. Peptide Synthesis

The fragment Ang_60–68_ including the amino acid sequence Ac-KNGNPHSEN-NH_2_ (molecular weight, MW, of 1105.5 g/mol, isoelectric point, PI, of 11.38), modified by N-terminal acetylation and the C-terminal amidation, was assembled by using the solid phase peptide synthesis strategy, on an initiator+ Alstra^TM^ fully automated microwave peptide synthesizer (Biotage, Uppsala, Sweden). The synthesis was performed on TGR resin (0.25 mmol/g) on 0.11 mmol scale using a 30 mL reactor vial. The coupling reactions were carried out by using 5-fold excess of amino acid, 5 equivalents of hydroxybenzotriazol/2-(1H-benzotriazole-1-yl)-1,1,3,3-tetramethylaminiumtetrafluoroborate/*N*,*N*-diisopropylethylamine (HOBt/TBTU/DIEA) in *N*,*N*-dimethylformamide (DMF), under mixing for 10 min at room temperature. Fmoc deprotection steps were performed at room temperature by using 20% of piperidine in DMF for 15 min. The N-terminal amino group was acetylated using a DMF solution containing acetic anhydride (6% *v*/*v*) and DIEA (5% *v*/*v*). The resin was washed with dichloromethane and dried on synthesizer. The peptide was purified by preparative reversed-phase chromatography (rp)-HPLC using a PrepStar 200 (model SD-1, Varian, Palo Alto, CA, USA) equipped with a Prostar photodiode array detector, with a protocol previously reported [48]. The peptide Ac-KNGNPHRENC-NH_2_ (Ang_60–68_Cys) was purchased from CASLO (Lyngby, Denmark). 

### 4.3. Protein Expression

The human angiogenin expression was carried out following the method reported by Holloway et al. (2001) [82]. Briefly, the *E. coli* (BL21(DE3)) expression strain was cultured at 37 °C under shaking (speed of 180 r.p.m.) in 5 mL of terrific broth (TB) (12 g peptone, 24 g granulated yeast extract, 4 mL glycerol 87%, 900 mL of distilled H_2_O) supplemented with ampicillin (100 μg/mL). After 24 hrs of incubation the whole volume of the bacterial culture was inoculated in 1000 mL of fresh broth. When the density of the culture had reached the OD_600 nm_ value of 0.8, the Ang expression was induced by the addition of 1 mM IPTG and the incubation was continued for additional 2 h. Afterwards, the cell culture was harvested by centrifugation (15 min at 1503 RCF) and cells were lysed with 30 mL of lysis buffer (50 mM Tris-HCl, 2 mM EDTA, pH = 8) by using a high-pressure homogenizer (Emulsiflex, Ottawa, Canada) and a sonication step (Sonicator Q700, Qsonica, Newtown, CT, USA). Lysate was centrifuged (40 min at 15,871 RCF) and the pellet was re-suspended in 25 mL of lysis buffer supplemented with 1% (*v*/*v*) Triton X-100. Sonication and centrifugation steps were repeated twice and the final pellet was dissolved in 30 mL of denaturation buffer (0.24 M GdnHCl, 100 mM Tris-HCl, 1 mM EDTA, 4 mM NaCl, 0.4 mM DTT).

The expressed recombinant angiogenin (rANG) was refolded from inclusion bodies according to the procedure described by Jang et al. [83] and then purified by a cation exchange chromatography performed on an automated chromatographic workstation (Akta prime, GE Healthcare, Milan, Italy) equipped with a 15 × 1.6 cm column packed with SP Sepharose Fast Flow (GE Healthcare, Milan, Italy). After a washing step with 25 mM Tris-HCl (pH = 8.0), rAng was eluted with 25 mM Tris-HCl, 1 M NaCl (pH = 8.0) buffer solution. Sodium dodecyl sulphate-polyacrylamide gel electrophoresis (SDS-PAGE) (10% bis-tris, Invitrogen, Carlsbad, CA, USA, 1 mm × 15 well) was carried out to evaluate the presence of dimers. 

To obtain wild-type angiogenin (ANG), rANG was incubated with 1 nM *Aeromonas* aminopeptidase, at the concentration of 1 × 10^−5^ M in 200 mM PBS (pH = 7.2) (overnight at 37 °C under gentle shaking). This procedure allows for the specific removal of the N-terminal methionine residue, Met(-1), in the primary sequence of rANG, thus obtaining the N-terminal glutamine residue, Glu1, that spontaneously cyclises to the pyroglutamate residue, PyrGlu1, which is characteristic of ‘native’ wtANG.

The reaction mixture was purified by dialysis (Spectra/por MWCO 6–8000) (Fisher Scientific, Hampton, NH, USA), which replaces PBS with 25 mM Tris-HCl (pH 7.4) buffer solution, followed by cation-exchange chromatography. The native folding of wtANG was evaluated by testing the ribonucleolytic activity of the protein, according to the procedure reported by Halloway et al. [82]. The protein concentration was determined by means of UV-visible spectroscopy (ε_280nm_ = 12,500 M^−1^ cm^−1^) [38].

### 4.4. UV–Visible Spectroscopy, Circular Dichroism Spectroscopy and Dynamic Light Scattering (DLS) Analyses

UV-visible spectra of the aqueous dispersions were measured on a Lambda 2S spectrometer (Perkin Elmer, Waltham, MA, USA) using conventional quartz cells (light path 1 cm and 0.1 cm) under the following conditions: bandwidth, 1 nm; scan rate, 100 nm/min; response, medium; data interval, 0.5 nm. Circular dichroism (CD) spectra in the 290–750 nm UV-visible region were recorded at 25 °C in a constant nitrogen flow on a model 810 spectropolarimeter (Jasco, Cremella (LC), Italy) equipped with a Xe lamp. The following conditions were used: scan rate, 50 nm min^−1^; bandwidth, 1 nm; scan rate, 50 nm/min; response, 4 s; accumulation, 3 times; data interval, 0.5 nm. Aqueous solution of (+)-ammonium camphorsulfonate-d_10_ (0.06%) was used for a calibration of the spectrometer sensitivity and wavelength (θ = 190.4 mdeg at λ = 290.5 nm).

For the hydrodynamic size determination, a dynamic light scattering (DLS) nanoparticle size analyser (LB-550, Horiba, Rome, Italy) was used. The instrument, equipped with temperature controller in the range of 5–70 °C, could detect particle size in the range of 1 nm–6 μm; response time, about 30 s. The results are presented as the mean of at least three measurements.

### 4.5. Synthesis and Functionalisation of Gold Nanoparticles

Gold nanoparticles were synthesized modifying the method pioneered by Turkevich. This method uses the chemical reduction of the chloroauric acid by the action of trisodium citrate that acts as both reducing and capping agent [84]. The synthesis was carried out as follows. All glassware was cleaned with aqua-regia rinsing (HCl:HNO_3_, 1:3 volume ratio) and then washed with MilliQ water immediately before starting the experiments. Gold(III) chloride dihydrate was dissolved in 20 mL of ultrapure Millipore water. The solution, at the final concentration of 1 mM, was heated to boiling on a hot plate while it is stirred in a 50 mL beaker. 2 mL of a 1% (*m*/*v*) solution of trisodium citrate dihydrate was quickly added to the rapidly-stirred auric solution. As soon as the solution turned from yellow to deep red, AuNPs were formed and the beaker was removed from the hot plate. 

The concentration of synthesized AuNPs was typically of 16 nM, as estimated by the UV–visible spectra, according to the molar extinction coefficient *ε* (in M^−1^cm^−1^,) calculated by the following equation [85]: $\varepsilon_{gold}=Ad^{\gamma }$, where *d* in (nm) is the core diameter of the nanoparticle, A and γ are constants (d ≤ 85 nm: $A=4.7\times 10^{4}$, γ=3.30; d > 85 nm: $A=1.6\times 10^{8}$, γ=1.47). 

To calculate *d*, the UV-visible parameters of the plasmon peak were used, according to the following equation [47]: $d=\frac{\lambda \max-515.04}{0.3647}$.

In order to remove the excess of sodium citrate, the citrate-capped gold nanoparticles were washed through two centrifugation steps (15 min at 6010 RCF), with rinsing in between and at the end with 3-(*N*-morpholino)propanesulfonic acid) -Tris(2-carboxyethyl)phosphine hydrochloride MOPS-TCEP buffer. To prepare the MOPS-TCEP buffer, 1 mM MOPS buffer solution (added with 0.27 mM KCl and 13.7 mM NaCl) was mixed to TCEP at 1:1 molar ratio, and the pH corrected to 7.4 (25 °C) by the addition of concentrated NaOH. 

The pellets of the rinsed citrate-capped gold nanoparticles were resuspended in 1 mM MOPS-TCEP at the concentration of 1.5×10^−8^ M, corresponding to 1.7 × 10^8^ AuNP/mL, as determined by the absorbance of the plasmon peak, and functionalized by physical adsorption (for Ang_60–68_ and ANG), and prevalent chemisorption (for Ang_60–68_Cys). The functionalization was carried out through the gradual addition, in a concentration range from 5 × 10^−6^ M up to 3 × 10^−5^ M, of the two different peptides (Ang_60–68_ and Ang_60–68_Cys) and through the one step addition of the whole protein (ANG) at the concentration of 1 × 10^−7^ M, to 1.5 × 10^−8^ M aqueous dispersion of AuNP and analysed by UV–visible spectroscopy titrations. Eventually, to rinse off unbounded or weakly bound biomolecules, the peptide-AuNP hybrid systems were purified by two centrifugation steps (15 min at 6010 RCF), with rinsing in between and at the end with MOPS-TCEP buffer. 

### 4.6. Cellular Experiments

Human neuroblastoma cells (SH-SY5Y cell line) were cultivated in full medium, i.e., DMEM/F12 supplemented with 10% FBS, 2 mM l-glutamine and 100 μg mL^−1^ streptomycin. For differentiation, cells were seeded at a density of 2.3 × 10^5^ cells/mL in full medium for 24 hrs and then neuronal differentiation of SH-SY5Y was induced by treatment for 5 days in vitro (DIV) with 10 µM of retinoic acid (RA) for 5 days in Dulbecco’s Modified Eagle Medium (DMEM) high glucose medium supplemented with 0.5% of FBS. 

Human glioblastoma cell line (A172) was cultivated in DMEM (n. 30–2002) supplemented with 10% FBS and 100 μg·mL^−1^ streptomycin. The cell cultures were grown in tissue-culture treated Corning^®^ flasks (Sigma-Aldrich) in humidified atmosphere (5% CO_2_) at 37 °C (HeraCell 150C incubator, Heraeus, Hanau, Germany). For the cellular treatments, the day before the experiment glioblastoma cells were seeded at a density of 2 × 10^5^ cells/mL in full medium. 

#### 4.6.1. Cellular Experiments

Corning^®^ 48 well multiwell plates were used for cytotoxicity assays (Sigma-Aldrich). The effect of AuNPs with ANG protein and Ang peptides on cell viability was tested at 50–60% of cell confluence by incubation with the compounds with concentrations 5 and 10 nM with or without 20 uM of copper for 24 hrs in DMEM medium supplemented with 0.5% of FBS. The viable cells were quantified by the reaction with MTT. After 90 min, the reaction was stopped by adding DMSO, and absorbance was measured at 570 nm (Varioskan^®^ Flash Spectral Scanning Multimode Readers, Thermo Scientific, Waltham, MA, USA). Results were expressed as % of viable cells over the concentration of each compound. The experiments were repeated at least five times in triplicate and results expressed as mean ± standard error of the mean (SEM). The statistical analysis was performed with a one-way Analysis of Variance (ANOVA test, by using the Origin software, version 8.6, Microcal, Northampton, MA, USA.

#### 4.6.2. Western Blot (WB) Analysis

For the determination of protein amount by WB, cells were cultivated at 37 °C (in 5% CO_2_ atmosphere) on Corning^®^ tissue-culture treated culture dishes 60 mm × 15 mm (D × H) (Sigma-Aldrich) at 80% of confluence. Cells lysates were prepared by cells treatment with RIPA lysis buffer (50 mM Tris-HCl, pH 8.0, 150 mM NaCl, 0.5 mM EDTA, 1% Triton X-100, 0.5 mM EGTA, 1% NP40) containing an inhibitor of the protease and phosphatase cocktail. Immediately after the addition of the buffer, cells were collected by the scratch method and transferred to Eppendorf tubes (1.5 mL of size, purchased from Sigma-Aldrich) for incubation on ice for 30 min. After a centrifugation step (10 min at 18,407 RCF) the supernatants were collected and the protein concentration was measured by Bradford’s method using BSA as the standard curve. SDS-PAGE with precast gel (4–20%, mini-PROTEAN, BioRad, Hercules, CA, USA) was used to separate proteins lysates or Ang protein and peptides. Nitrocellulose membranes (Sigma-Aldrich) were used to transfer proteins from the gel. Membranes were incubated with blocking buffer (0.1% Tween20 in tris-buffered saline added with either 5% non-fat milk) at room temperature for 1 h, and then incubated with primary anti-angiogenin or anti-GAPDH antibody (code: sc-9044, 1:500 dilution, Santa Cruz Biotechnology (Dallas, TX, USA) or code: ab8245, 1:2000 dilution, Abcam (Cambridge, UK), respectively) overnight at 4 °C. After that, 1 h treatment with goat anti-rabbit or anti-mouse IgG horseradish peroxidase-conjugated secondary antibodies (code: AP307P and AP181P, respectively, 1:3000 dilution, MD Millipore Bioscience (Burlington, MA, USA). Measurements were performed by a ChemiDoc MP Imaging System (BioRad, Hercules, CA, USA) using enhanced Western Lighting Chemiluminescence Reagent Plus (PerkinElmer, Waltham, MA, USA). 

#### 4.6.3. Dot Blot Analysis

Peptides or whole protein were dissolved in phosphate buffer saline solution (PBS, pH = 7.4) at 0.2 mg/mL. Using narrow-mouth pipette tip, 2 µL of samples were spotted onto the nitrocellulose membrane. To block the non-specific sites membrane was incubated in 5% non-fat milk in 0.1% Tween20 in tris-buffered (30 min, room temperature). After that, membranes were incubated with primary anti-angiogenin antibody (code: sc-9044, 1:500 dilution, from Santa Cruz Biotechnology, Dallas, TX, USA) overnight at 4 °C. Than membranes were incubated with secondary antibody conjugated with horseradish peroxidase (HRP) enzyme (code: AP307P, 1:3000 dilution, MD Millipore Bioscience, Burlington, MA, USA) for 1 h. Measurements were performed by a ChemiDoc MP Imaging System (BioRad, Hercules, CA, USA) using enhanced Western Lighting Chemiluminescence Reagent Plus (PerkinElmer, Waltham, MA, USA).

#### 4.6.4. Quantitative qPCR

For qPCR experiments, cells were cultivated at 37 °C (in 5% CO_2_ atmosphere) on Corning^®^ tissue-culture treated culture dishes 60 mm × 15 mm (D × H) (Sigma-Aldrich) at 80% of confluence. Total RNA was extracted with TRIzol (Life Technologies, Foster City, CA, USA), according to the manufacturer’s instructions. RNA quantification was performed by Epoch™ Microplate Spectrophotometer (BioTek^®^, Winooski, VT, USA). Extracted RNA was reverse transcribed by using a High Capacity RNA-to-cDNA Kit (Life Technologies), according to the manufacturer’s instructions. Resulting cDNAs (30 ng per sample) were amplified through a LightCycler^®^ 480 System (Roche, Pleasanton, CA, USA). Single-gene specific assays were performed through real-time PCR by using Fast SYBR Green Master Mix (Life Technologies, Carlsbad, CA, USA) according to the manufacturer’s instruction. To allow statistical analysis, PCRs were performed in three independent biological replicates. 18S was used as housekeeping gene to normalize PCR data. Primer sequences are listed in Table 3.

#### 4.6.5. Sandwich ELISA Assay

Medium samples were collected after a 24-h treatment exposure with AuNPs, ANG and peptides, and hybrids Ang_60–68__NP, Ang_60–68_Cys_NP and ANG_NP in DMEM medium supplemented with 0.5% of FBS and centrifuged (14,000× *g*, 10 min), the supernatants were transferred into clean microtubes and stored at −80 °C until analysed. The concentration of VEGF release was determined from the cell culture media samples using ELISA sandwich assay. Polyvinyl chloride (PVC) microtiter plates were coated overnight at 4 °C with 5 µg/mL of capture antibody (anti-VEGF, code: PAB12284) in carbonate/bicarbonate buffer (pH 9.6). Then plates were washed twice with PBS, blocked by blocking buffer (5% non-fat dry milk/PBS) for 2 hrs at room temperature, washed with PBS and incubated with cell culture media samples for 90 min at 37 °C. After plates were washed with PBS, incubated for 2 hrs with 1 µg/mL of detection antibody (anti-VEGF, code: H00007422-M05), washed again, then incubated for 2 hrs with HRP-conjugated secondary antibody and washed with PBS. Result was detected by 3,3′,5,5′-tetramethylbenzidine (TMB) solution after the incubation for 15 min. The reaction was stopped by stopping solution (2 M H_2_SO_4_) and the optical density was measured at 450 nm by a plate reader (Varioskan^®^ Flash Spectral Scanning Multimode Reader, Waltham, MA, USA).

#### 4.6.6. Laser Scanning Confocal Microscopy (LSM)

SH-SY5Y cells were seeded (30 × 10^3^ cells per dish) and differentiated (see 2.6.1) in glass bottom dishes with 22 mm of glass diameter (WillCo-dish^®^, Willco Wells, B.V., Amsterdam Netherlands). Glioblastoma cells were seeded at the density 30 × 10^3^ cells per dish in glass bottom dishes with complete medium for 24 hrs until cellular adhesion with a minimal cell confluence of 50% was attained. Thereafter, cells were treated with AuNPs (5 nM) or ANG (100 nM) or angiogenin peptides (30 µM) and their hybrids in the presence or absence of copper for 2 hrs in DMEM high glucose medium without FBS. After the incubation time, cells were stained with nuclear dye Hoechst33342, washed with PBS, and fixed with high purity 2% paraformaldehyde in PBS (pH = 7.3). Afterwards, cells were permeabilized with 0.5% Triton X-100 with 0.1% BSA and stained firstly with a high-affinity F-actin probe (Actin Green 488 Ready Probes Reagent, ThermoFisher), conjugated to green-fluorescent Alexa Fluor^®^ 488 dye for 30 min, washed with PBS and then with anti-angiogenin antibody (code: sc-9044, 1:50 dilution, Santa Cruz Biotechnology, Dallas, TX, USA) overnight at 4 °C. After that, 1 h treatment with donkey anti-rabbit IgG H&L (Alexa Fluor^®^ 568) pre-adsorbed secondary antibodies (code: ab175692, 1:1000 dilution, MD Millipore Bioscience, Burlington, MA, USA). 

For multichannel imaging, fluorescent dyes were imaged sequentially to eliminate cross talk between the channels, namely: (i) the blue (ex405/em 425–475), for the emission of the Hoechst33342-stained nuclei, (ii) the green (ex488/em 500–530), for the emission of the Actin Green 488 Ready Probes Reagent, (iii) the red (ex543/em 560–700), for Alexa Fluor^®^ 568 of secondary antibody.

Confocal imaging microscopy was performed with a FV1000 confocal laser scanning microscope (LSM, Olympus, Tokyo, Japan), equipped with diode UV (405 nm, 50 mW), multiline Argon (457 nm, 488 nm, 515 nm, total 30 mW), HeNe(G) (543 nm, 1 mW) and HeNe(R) (633 nm, 1 mW) lasers. An oil immersion objective (60xO PLAPO) and spectral filtering system were used. The detector gain was fixed at a constant value and images were taken, in sequential mode, for all the samples at random locations throughout the area of the well. The image analysis was carried out using Huygens Essential software (by Scientific Volume Imaging B.V., Hilversum, The Netherlands).

## 5. Conclusions

The results obtained from experiment performed on A172 cells indicate that Ang_60–68_Cys and Ang_60–68_ anchored on AuNPs could be considered as effective inhibitors of glioblastoma tumour cell proliferation and VEGF release. However, the mechanism of action and potential side effects need to be elucidated further. Moreover, exogenous delivery of angiogenin by gold nanoparticles could represent a strategic approach to re-establish the physiological concentrations of angiogenin in the course of diseases in which the protein levels are strongly reduced and suggests that further studies are required to translate these effects into meaningful therapies.

Our results demonstrated that copper induced a decrease of VEGF mRNA transcription on d-SH-SY5Y cells. These data confirmed those of other studies, demonstrating the toxic effect of the copper on SH-SY5Y cells, particularly on the mitochondria, with decreased levels of mitochondrial proteins [86]. On the other hand, the treatment of A172 cancer cells with Cu(II) induced an increase of VEGF release, demonstrating the different role of the copper in both tumoral and non-tumoral cells. It has been demonstrated that copper induces the expression of VEGF in breast and hepatic cancer cells through the activation of the epidermal growth factor receptor/extracellular signal-regulated protein kinases (EGFR/ERK)/c-fos transduction pathway [87]. Surprisingly, the incubation of A172 cells with copper in presence of Ang_60–68_Cys_NP potentiated the inhibitory effect of the protein fragment on VEGF release, demonstrating the therapeutic potential of copper chelating agents against tumour progression. However, the mechanism of action and potential side effects need to be elucidated further. 

## Figures and Tables

**Figure 1 cancers-11-01322-f001:**
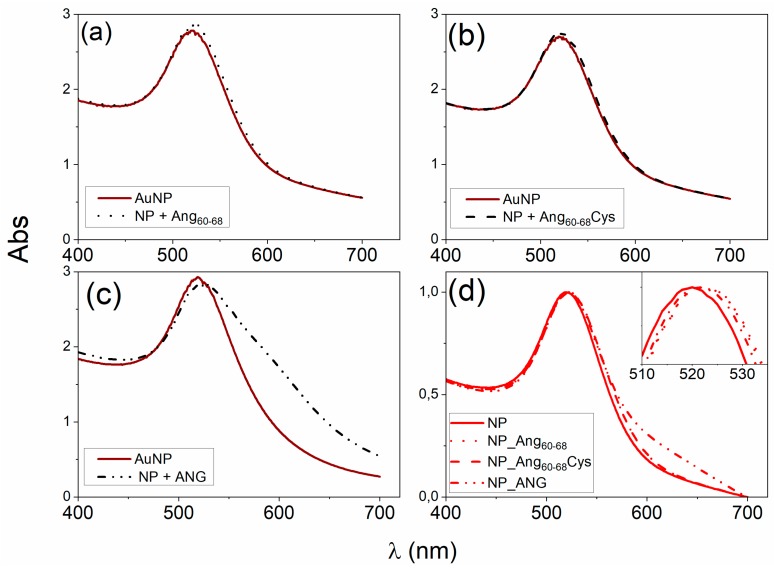
(**a**–**c**) Ultraviolet (UV)-visible spectra of gold nanopartilces (AuNPs) in the 1 mM 3-(N-morpholino)propanesulfonic acid)-Tris(2-carboxyethyl)phosphine hydrochloride (MOPS-TCEP) buffer (1:1 mol ratio) before and after the addition of: (**a**) 30 μM Ang_60–68_, (**b**) 30 μM Ang_60–68_Cys; (**c**) 100 nM angiogenin (ANG). (**d**) UV-visible spectra of the pellets collected after two rinsing steps by centrifugation (15 min at 6010 relative centrifugal force, RCF) and re-suspension in 1 mM MOPS-TCEP buffer.

**Figure 2 cancers-11-01322-f002:**
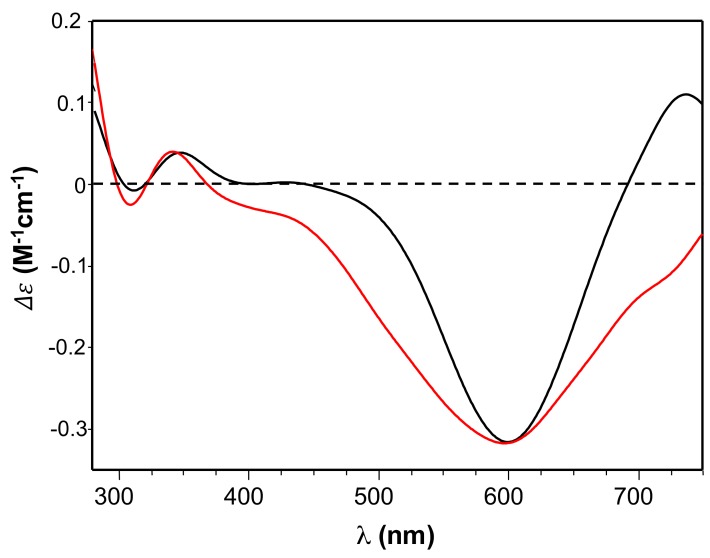
Circular dichroism (CD) spectra of Ang_60–68_ + CuSO_4_ (black line) and Ang_60–68_Cys + CuSO_4_ (red line) at pH = 7.4. Equimolar concentration of peptide and copper were used: [peptide] = [Cu(II)]= 1 × 10^−3^ M.

**Figure 3 cancers-11-01322-f003:**
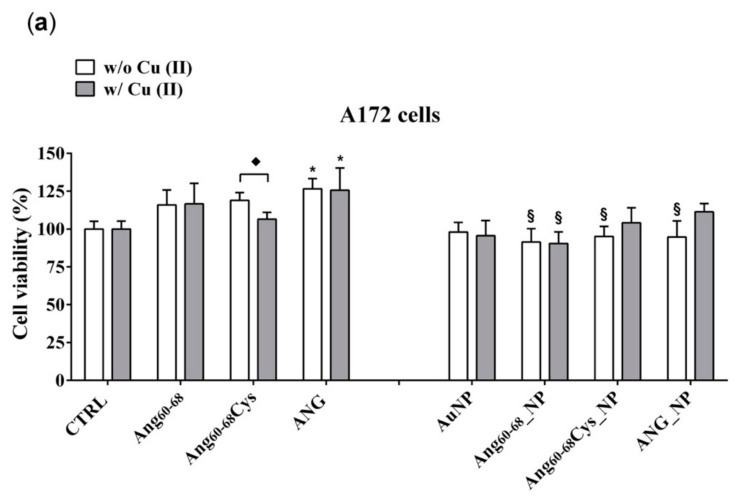
Cell viability determined by 3-(4, 5-dimethylthiazolyl-2)-2, 5-diphenyltetrazolium bromide (MTT) assay of A172 (**a**) and d-SH-SY5Y (**b**) cell lines. Cells were grown in basal culture medium (control: CTRL) and in culture medium supplemented with: Ang_60–68_ (30 μM), Ang_60–68_Cys (30 μM); ANG (100 nM), AuNP (9.4 nM = 1.4 × 10^8^ NP/mL), Ang_60–68__NP (1.4 nM = 4.0 × 10^6^ NP/mL, [Ang_60–68_] = 2.8 × 10^−12^ M), Ang_60–68_Cys_NP (1.4 nM = 4.0 × 10^6^ NP/mL, [Ang_60–68_Cys] = 2.6 × 10^−12^ M), ANG_NP (1.2 nM = 3.4 × 10^6^ NP/mL, [ANG]= 0.2 × 10^−12^ M). All conditions were evaluated in presence or absence of metal ions (copper sulphate: Cu(II), 20 μM). The bars represent means ± SD of three independent experiments performed in triplicate (S.D. = standard deviation). Statistically significant differences, determined by one-way analysis of variance ANOVA are indicated: * *p* ≤ 0.05 versus CTRL; ^§^
*p* ≤ 0.05 versus the respective treatment with free peptides/protein; ^♦^
*p* ≤ 0.05 versus the respective treatment w/o Cu(II).

**Figure 4 cancers-11-01322-f004:**
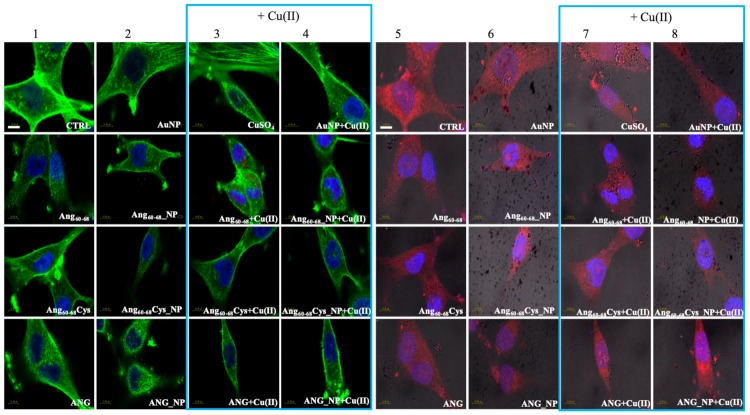
Confocal micrographs of A172 cells. Actin Green^®^488 (in green, ex/em= 488/500–530 nm) and Hoechst33342 (in blue, ex/em=405/425–475 nm) were used as F-actin and nuclear markers, respectively. Antibody against angiogenin shows angiogenin localisation in red (ex/em=543/560–700 nm) and micrographs are merged with optical bright field images (in grey). Before treatments, cell were rinsed with fresh culture medium and incubated for 2 h with basal culture medium (control: CTRL) and in culture medium supplemented with: Ang_60–68_ (30 μM), Ang_60–68_Cys (30 μM); ANG (100 nM), AuNP (9.4 nM = 1.4 × 10^8^ NP/mL), Ang_60–68__NP (1.4 nM = 4.0 × 10^6^ NP/mL, [Ang_60–68_] = 2.8 × 10^−12^ M), Ang_60–68_Cys_NP (1.4 nM = 4.0 × 10^6^ NP/mL, [Ang_60–68_Cys] = 2.6 × 10^−12^ M), ANG_NP (1.2 nM = 3.4 × 10^6^ NP/mL, [ANG]= 0.2 × 10^−12^ M). Scale bar = 10 μm.

**Figure 5 cancers-11-01322-f005:**
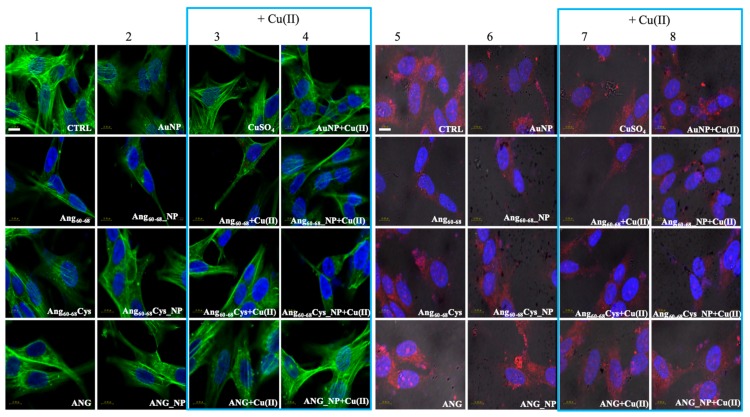
Confocal micrographs of d-SH-SY5Y cells Actin Green^®^488 (in green, ex/em = 488/500–530 nm) and Hoechst33342 (in blue, ex/em = 405/425–475 nm) were used as F-actin and nuclear markers, respectively. Antibody against angiogenin shows angiogenin localisation in red (ex/em=543/560–700 nm) and micrographs are merged with optical bright field images (in grey). Before treatments, cell were rinsed with fresh culture medium and incubated for 2 h with basal culture medium (control: CTRL) and in culture medium supplemented with: Ang_60–68_ (30 μM), Ang_60–68_Cys (30 μM); ANG (100 nM), AuNP (9.4 nM = 1.4 × 10^8^ NP/mL), Ang_60–68__NP (1.4 nM = 4.0 × 10^6^ NP/mL, [Ang_60–68_] = 2.8 × 10^−12^ M), Ang_60–68_Cys_NP (1.4 nM = 4.0 × 10^6^ NP/mL, [Ang_60–68_Cys] = 2.6 × 10^−12^ M), ANG_NP (1.2 nM = 3.4 × 10^6^ NP/mL, [ANG]= 0.2 × 10^−12^ M). Scale bar = 10 μm.

**Figure 6 cancers-11-01322-f006:**
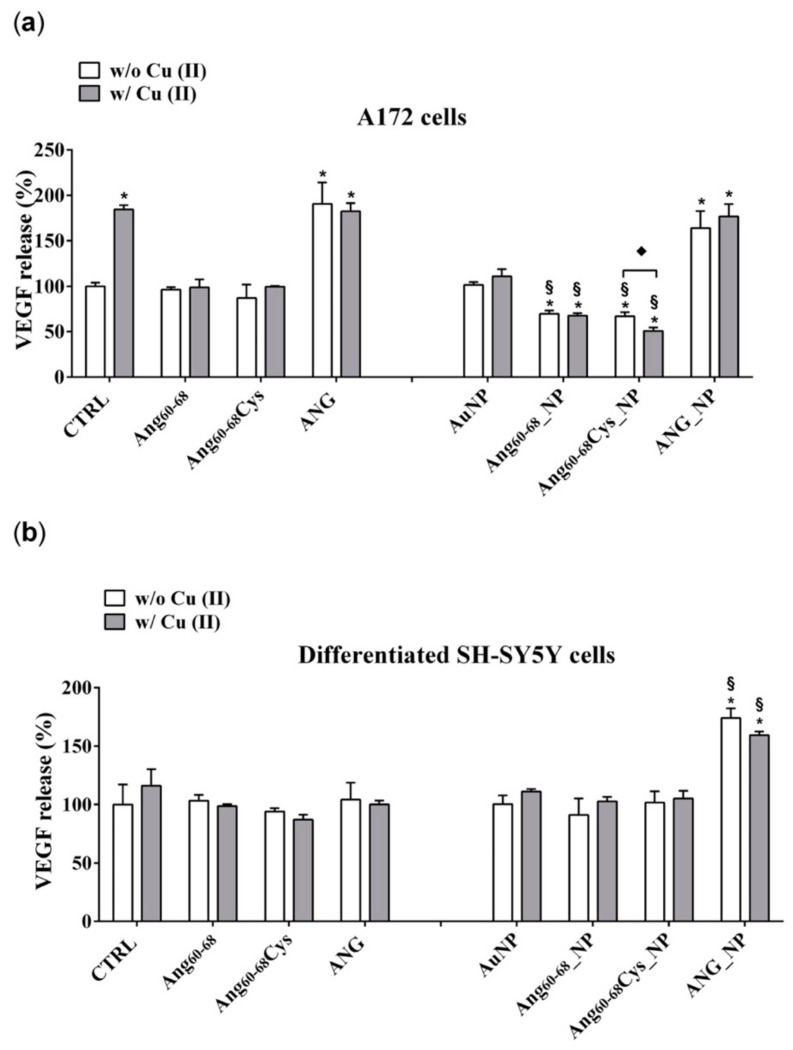
Vascular endothelial growth factor (VEGF) release in the medium of confluent cultures of A172 (**a**) and differentiated d-SH-SY5Y cells (**b**). Cells were grown in basal culture medium (control: CTRL) and in culture medium supplemented with: Ang_60–68_ (30 μM), Ang_60–68_Cys (30 μM); ANG (100 nM), AuNP (9.4 nM = 1.4 × 10^8^ NP/mL), Ang_60–68__NP (1.4 nM = 4.0 × 10^6^ NP/mL, [Ang_60–68_] = 2.8 × 10^−12^ M), Ang_60–68_Cys_NP (1.4 nM = 4.0 × 10^6^ NP/mL, [Ang_60–68_Cys] = 2.6 × 10^−12^ M), ANG_NP (1.2 nM = 3.4 × 10^6^ NP/mL, [ANG]= 0.2 × 10^−12^ M). All conditions were evaluated in presence or absence of metal ions (copper sulphate: Cu(II), 20 μM). Sandwich enzyme-linked immunosorbent assay (ELISA) with monoclonal anti-VEGF antibody was used. The bars represent means ± SD of three independent experiments performed in triplicate (S.D. = standard deviation). Statistically significant differences, determined by one-way analysis of variance ANOVA are indicated: * *p* ≤ 0.05 versus CTRL; ^§^
*p* ≤ 0.05 versus the respective treatment with free peptides/protein; ^♦^
*p* ≤ 0.05 versus the same treatment w/o Cu (II).

**Figure 7 cancers-11-01322-f007:**
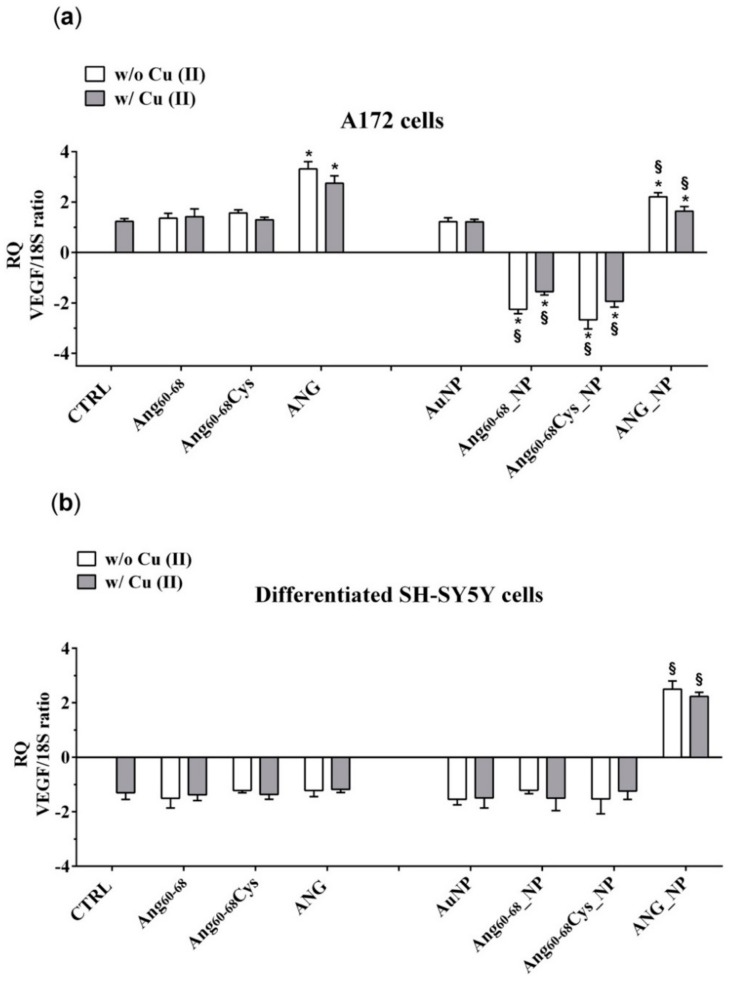
Vascular endothelial growth factor (VEGF) mRNA levels determination by qPCR in A172 (**a**) and differentiated d-SH-SY5Y cells (**b**). Cells were grown in basal culture medium (control: CTRL) and in culture medium supplemented with: Ang_60–68_ (30 μM), Ang_60–68_Cys (30 μM); ANG (100 nM), AuNP (9.4 nM = 1.4 × 10^8^ NP/mL), Ang_60–68__NP (1.4 nM = 4.0 × 10^6^ NP/mL, [Ang_60–68_] = 2.8 × 10^−12^ M), Ang_60–68_Cys_NP (1.4 nM = 4.0 × 10^6^ NP/mL, [Ang_60–68_Cys] = 2.6 × 10^−12^ M), ANG_NP (1.2 nM = 3.4 × 10^6^ NP/mL, [ANG]= 0.2 × 10^−12^ M). All conditions were evaluated in presence or absence of metal ions (copper sulphate: Cu(II), 20 μM). Relative quantification is referred to untreated cells (CTRL). Data normalized with respect to the expression level of S18 mRNA. The bars represent means ± standard deviation (SD) of three independent experiments performed in triplicate. Statistically significant differences, determined by one-way analysis of variance (ANOVA) are indicated: * *p* ≤ 0.05 versus control; ^§^
*p* ≤ 0.05 versus the respective treatment with free peptides/protein.

**Table 1 cancers-11-01322-t001:** Hydrodynamic size of the different NPs either before or after the functionalisation with the peptides or the protein, and after the addition of 20 μM CuSO_4_ aqueous solution.

	Hydrodynamic Size (nm)		
Peptide/Protein	Nanoparticle (NP) + Peptide/Protein ^(1)^	Peptide/Protein_NP ^(2)^	Peptide/Protein_NP + Cu(II) ^(3)^
-	29 ± 3	30 ± 2	31 ± 5
Ang_60–68_	28 ± 4	37 ± 4	175 ± 10
Ang_60–68_Cys	30 ± 3	29 ± 5	281 ± 42
ANG	53 ± 4	41 ± 6	47 ± 5

^1^ peptide/protein-added NP samples; ^2^ peptide/protein_NP samples after the centrifugation and rinsing steps; ^3^ same pellets as 2 added with copper ions.

**Table 2 cancers-11-01322-t002:** Protein fraction shell value (g) and peptide/protein coverage (Γ) calculated from the changes in the wavelength of maximum absorption ($\lambda_{max}$) of AuNP plasmon peak for peptide/protein added nanoparticles. The ideal monolayer coverage of the peptide/protein in the end-on and side-on limit configuration are given for comparison.

Sample	*g* ^(1)^	*Γ* (ng/cm^2^)	*Γ*^(2)^ (molecules/NP)	Ideal Monolayer Coverage ^(3)^ (molecules/NP)	
				End-on	Side-on
Ang_60–68__NP	0.74	79	194	177	83
Ang_60–68_Cys_NP	0.74	79	178	202	149
ANG_NP	0.90	165	32	23	10

^1^ Values calculated from Equation (1) in Materials and Methods by considering the refractive index values (n) at 550 nm of 1.335 for water [60] and 1.38 for a pure protein [59], respectively. ^2^ Values calculated from Equation (3) in Materials and Methods, given the molecular weights (MW) of 1105.5 g/mol for Ang_60–68_, 1209.3 g/mol for Ang_60–68_Cys and 14,200 g/mol for ANG, respectively. ^3^ Calculated by using the average molecular dimensions (in nm^3^) respectively of (1.7 × 1.5 × 3.2) for Ang_60–68_, (1.6 × 1.4 × 1.9) for CysAng_60–68_ [44], and (7 × 6.2 × 3.2) for ANG [61].

**Table 3 cancers-11-01322-t003:** Primer sequences for vascular endothelial growth factor A (VEGFA) and 18S ribosomal RNA (18S) genes.

Gene Symbol	Forward	Reverse
Human VEGFA	5′-ATCTTCAAGCCATCCTGTGTGC-3′	5′- GAGGTTTGATCCGCATAATCTG-3′
18S	5′-AGTCCCTGCCCTTTGTACACA-3′	5′-GATCCGAGGGCCTCACTAAAC-3′

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
