# Peer review of "A Tunable Nanoplatform of Nanogold Functionalised with Angiogenin Peptides for Anti-Angiogenic Therapy of Brain Tumours"

_cancers, 2019, doi:10.3390/cancers11091322_

Round 1

Reviewer 1 Report

The article entitled “A Tunable Nanoplatform of Nanogold Functionalised with Angiogenin Peptides for Anti- Angiogenic Therapy of Brain Tumours” by Irina Naletova et al.  The reported work discusses the use of spherical gold nanoparticles of a size 10 & 30 nm were functionalized with the peptide fragment from the putative cell membrane binding domain.   Cellular treatments and investigation on two brain cell lines were investigated. Cell viability, cytoskeleton, angiogenin translocation and VEGF release. some of the issue listed below still must be looked at.

Broad comments

Please define the abbreviations first time they appear in the text for example VEGF line 31 and overall the whole manuscript “the wavelength at the maximum absorbance (max = 519 nm) and the full width at half maximum (FWHM = 54 nm) point to the formation of a monodisperse gold colloidal solution of spherical nanoparticles” lines 121 and 122. The lambda max would be indicative of spherical nanoparticles formation but by no means it can indicate a monodisperse formation of AuNPs. You would need dynamic light scattering (DLS) and its polydispersity index to verify this claim “at the concentration of 3·10-5 M for both peptides, induces comparable red-shifts (Δλmax = 3 nm)” line 125; How was the concentration obtained and is it for the peptide or for the AuNPs “The addition of the whole protein ANG (Fig. 1c), at the concentration of 1·10-7 M,” Line 128 Please include the excitation coefficient used for calculating the concentration or verify the measurements method “centrifugation (8,000 r.p.m,” please report the speed in g forces “The hydrodynamic size of ~ 30 nm for the aqueous dispersion of gold nanoparticles (1.7·108 NP/mL)” line 147. If you check the reported size in line 124 the particles size changed from 11 nm to 30 nm. Please comment on this? “where a fraction of loosely bound proteins molecules was therefore likely rinsed off by the washing steps of the protein-nanoparticle hybrids” line 154, how loosely bound protein molecules were determined? “unchanged for bare AuNP, instead a dramatic increase in the hydrodynamic diameter was found for Ang60-68_NP (to ~ 180 nm) and Ang60-68Cys_NP (to ~300 nm), respectively.” Line 157, the reported particles size is out of the under 100 nm particles therefore their must be reported as submicron particles “before and after addition of 20 M CuSO4” is this aqueous solution Please revise the table legend. Either before or after and combine the two comments in one legend Please revise the spaces between the numerical values and the measurements unit

Specific Comments

The introduction doesn’t capture the massive number of naturally occurring nanoparticles and their roles in cancer targeting. Its merely focused on the ANG discovery and role which is important but should include, protein cages, polymeric, micelles and liposomes for selective targeting of cancer in vivo and in vitro. Please revise accordingly. Upon the incubation of the peptide sequences Ang60-68 and Ang60-117 68Cys with the AuNPs. Was there any purification steps to ensure that the peptide is conjugated to the surface of the nanoparticles to ensure no free peptides remains unbound in the samples There is a discrepancy in reported particles size from UV-vis spectrum and the DLS. The hydrodynamic measurements is more reliable and widely used in the nanoscience field for reporting the size. Please use it as the main size measurements tool “was largely increased in comparison to the bare AuNP, thus suggesting nanoparticle aggregation prompted by the presence of the protein at the nanoparticle surface” line 150. This is confusing; the main purpose of nanoparticles functionalization is to increase the stability of the particles and reduce the aggregation. Please revise? Sometimes the author uses Fig others Figure; please be consistent Figure 4 & 5 the scale bar is not clear on the images The zeta potential of all generated NPs should be included as it will show that the peptide is conjugated to the surface of the particles. The reported nanoparticles count that were incubated with the cell lines. Could the authors clarify the source of these particles count? Please refer to the CAS number of the chemicals reported in this manuscript. What is the confluency of the cells upon conducting experiment? Please report the centrifugal forces in (g) not rpm “and pH was corrected to 7.4 (25 °C).” how was the pH corrected Please include all the chemicals from the same company together in section 4.1. please revise accordingly Authors should use the spectrophotometric technique to quantify the AuNPs content of NPs. So, the gold concentration will be reported. For example (Zuber, A.; Purdey, M.; Schartner, E.; Forbes, C.; van der Hoek, B.; Giles, D.; Abell, A.; Monro, T.; Ebendorff-Heidepriem, H. Detection of gold nanoparticles with different sizes using absorption and fluorescence-based method. Sens. Actuators B 2016, 227, 117–127). Alternatively, elemental analysis techniques should be reported Please report the operating conditions of instruments/size, operating voltage, angles and other relevant technical instrumental settings (DLS and DC, UV-vis)? Authors should include TEM images of the particles both modified with the peptide and bare Some of the references lacks volume, issue or pages, please revise

Reviewer 2 Report

Dear author,

I have read the article entitled, ' A Tunable Nanoplatform of Nanogold
3 Functionalised with Angiogenin Peptides for Anti4 Angiogenic Therapy of Brain Tumours' with high interest. The article is novel and experiments are well carried out. Hence, I recommend to accept the manuscript after minor revisions.

Authors need more characterization data to support the binding of peptide and protein to gold nanoparticles especially FTIR and Zeta potential. How much is the binding efficacy of peptide/protein to gold nanoparticles?

Overall, the article is well written, but minor grammatical and English corrections need throughout the manuscript.

Round 2

Reviewer 1 Report

No Comments